# MicroRNA Profiles in Intestinal Epithelial Cells in a Mouse Model of Sepsis

**DOI:** 10.3390/cells12050726

**Published:** 2023-02-24

**Authors:** Siqingaowa Caidengbate, Yuichi Akama, Anik Banerjee, Khwanchanok Mokmued, Eiji Kawamoto, Arong Gaowa, Louise D. McCullough, Motomu Shimaoka, Juneyoung Lee, Eun Jeong Park

**Affiliations:** 1Department of Molecular Pathobiology and Cell Adhesion Biology, Mie University Graduate School of Medicine, 2-174 Edobashi, Tsu, Mie 514-8507, Japan; 2Department of Emergency and Disaster Medicine, Mie University Graduate School of Medicine, 2-174 Edobashi, Tsu, Mie 514-8507, Japan; 3Department of Neurology, McGovern Medical School, The University of Texas Health Science Center at Houston, 6431 Fannin Street, Houston, TX 77030, USA; 4UTHealth Houston Graduate School of Biomedical Sciences, The University of Texas MD Anderson Cancer Center, 6767 Bertner Avenue, Houston, TX 77030, USA

**Keywords:** sepsis, cecal slurry injection, inflammation, intestinal epithelial cell, miRNA

## Abstract

Sepsis is a systemic inflammatory disorder that leads to the dysfunction of multiple organs. In the intestine, the deregulation of the epithelial barrier contributes to the development of sepsis by triggering continuous exposure to harmful factors. However, sepsis-induced epigenetic changes in gene-regulation networks within intestinal epithelial cells (IECs) remain unexplored. In this study, we analyzed the expression profile of microRNAs (miRNAs) in IECs isolated from a mouse model of sepsis generated via cecal slurry injection. Among 239 miRNAs, 14 miRNAs were upregulated, and 9 miRNAs were downregulated in the IECs by sepsis. Upregulated miRNAs in IECs from septic mice, particularly miR-149-5p, miR-466q, miR-495, and miR-511-3p, were seen to exhibit complex and global effects on gene regulation networks. Interestingly, miR-511-3p has emerged as a diagnostic marker in this sepsis model due to its increase in blood in addition to IECs. As expected, mRNAs in the IECs were remarkably altered by sepsis; specifically, 2248 mRNAs were decreased, while 612 mRNAs were increased. This quantitative bias may be possibly derived, at least partly, from the direct effects of the sepsis-increased miRNAs on the comprehensive expression of mRNAs. Thus, current in silico data indicate that there are dynamic regulatory responses of miRNAs to sepsis in IECs. In addition, the miRNAs that were increased with sepsis had enriched downstream pathways including Wnt signaling, which is associated with wound healing, and FGF/FGFR signaling, which has been linked to chronic inflammation and fibrosis. These modifications in miRNA networks in IECs may lead to both pro- and anti-inflammatory effects in sepsis. The four miRNAs discovered above were shown to putatively target *LOX*, *PTCH1*, *COL22A1*, *FOXO1*, or *HMGA2*, via in silico analysis, which were associated with Wnt or inflammatory pathways and selected for further study. The expressions of these target genes were downregulated in sepsis IECs, possibly through posttranscriptional modifications of these miRNAs. Taken together, our study suggests that IECs display a distinctive miRNA profile which is capable of comprehensively and functionally reshaping the IEC-specific mRNA landscape in a sepsis model.

## 1. Introduction

Sepsis is a leading cause of global mortality. Epidemiological studies have shown that the mortality of the patients in intensive care units with sepsis is higher than 40% [1]. Among the 49 million people who are affected annually worldwide, approximately 11 million individuals die [2]. Multiple organ dysfunction (MOD) is a pathologic condition which contributes to the increase in morbidity and mortality in sepsis [3,4,5]. The aberrant host response to polymicrobial infection and inflammation is the leading cause of MOD [6,7]. Sepsis has become increasingly recognized as a condition that promotes an overactive host immune response followed by MOD [6,8]. The pathophysiology of sepsis development is immunologically and spatiotemporally complex. Upregulation of both pro- and anti-inflammatory responses occurs during initial stages of infection, followed by morbid outcomes which are associated with hyperinflammation or immune paralysis [9,10].

The intestines are sensitive to sepsis-induced inflammation. Splanchnic ischemia and mucosal injury occur in the intestines upon onset of sepsis [11]. The injured mucosa upregulates and secretes pro-inflammatory mediators into the systemic vasculature, inducing systemic inflammation in multiple organs, including the brain [12]. Intestinal epithelial cells (IECs) constitute a single-layered lining that plays an important role in host defense by providing a physical barrier between the luminal surface containing microbe-derived factors and the host. These cells transfer signals bidirectionally between the host and microbes to mount appropriate immune responses [13,14,15,16,17]. Intrinsic and extrinsic inflammatory stimuli induced by sepsis disrupt the intestinal barrier and enhance epithelial permeability, resulting in the development of systemic inflammatory response and MOD [18,19]. IECs are one of the key players that regulate immune pathophysiology in sepsis [20,21]. Thus, understanding the IEC response to sepsis is highly significant. We investigated changes in the expression profiles of epigenetic regulators in septic IECs, such as small regulatory RNAs (e.g., microRNAs; miRNAs). Such approaches may be helpful to better understand how IECs reshape their post-sepsis gene expression and mediate changes in downstream signaling pathways.

Cecal slurry (CS) injection is a widely accepted model to induce chronic polymicrobial sepsis. In this model, cecal contents of other animals are administered into the peritoneal cavity of recipient mice, as described previously [22,23,24,25,26]. This model has been used to establish experimental sepsis in neonatal mice using freshly prepared samples [22,24]. The CS-injection model can induce differential mortality in a dose-dependent manner and is dependent more on bacterial infection than an endotoxin effect [27,28]. Starr et al. have also improved the method of CS preservation via maintaining bacterial viability in samples for at least several months [27].

In this study, we used the CS-injection model to induce chronic sepsis. We analyzed miRNA profiles specifically in IECs. IECs isolated from CS-injected groups dynamically responded to sepsis by altering their miRNA profiles. Subsequent in silico analysis showed that the miRNAs upregulated in IECs after sepsis regulate both pro- and anti-inflammatory downstream pathways, activating pathways related to protective and detrimental effects of epithelial inflammation.

## 2. Materials and Methods

### 2.1. Mice

C57BL/6J (13–15w-old male) mice were purchased from Japan SLC (Shizuoka, Japan). The mice were maintained in the Mie University Experimental Animal Facility at a specific pathogen-free condition under a 12-h light–dark cycle. The mice were given water and food ad libitum. All the experiments were conducted according to protocols approved by the Ethics Review Committee for Animal Experimentation of Mie University (approval number: #2019-41-1).

### 2.2. Polymicrobial Sepsis Induction

Polymicrobial sepsis was induced by intraperitoneally injecting CS, as previously described [25,27]. In brief, 0.25 mL of CS resuspended in 10% glycerol/PBS was injected into each mouse. Mice in the sham cohort were given the same volume of 10% glycerol intraperitoneally. Twelve hours after CS injection, mice in both CS and sham cohorts were intraperitoneally injected with antibiotics of 3 mg meropenem (Wako, Osaka, Japan) and 3 mg cilastatin (Wako) per mouse, seven times at twelve-hour intervals. All mice were subcutaneously injected with 0.7 mL 0.9% saline. Previously, the survival rate observation and sampling were conducted from days 14 to 30 [25] and from days 15 to 17 after CS injection (an unpublished report by Akama et al.), respectively. Thus, we used day 17 after CS injection in the current model, as previously described, with slight modifications to provide the mice within same cohorts with similar sepsis conditions.

### 2.3. IEC Isolation and Enrichment

IECs were isolated as previously described [29,30,31] with slight modifications. In brief, small intestines were collected from mice after euthanasia and Peyer’s patches, mesentery, and fats were removed prior to further processing. The tissues were opened and washed with ice-cold RPMI-1640 (Nacalai, Kyoto, Japan). The rinsed tissues were cut into small pieces at 1-cm length and incubated in RPMI-1640 containing 10% FBS (Equitech-Bio, Kerrville, TX, USA) and 2 mM ethylenediaminetetraacetic acid (EDTA) (Wako) for 30 min at 37 °C. The digested tissues were filtered using a 70-μm cell strainers (Corning, Glendale, AZ, USA). The filtered cell suspension was resuspended in 40% Percoll (GE Healthcare Life Sciences, Chicago, IL, USA) and applied to gradients of 25, 40, and 75% Percoll. After centrifugation in AX-511 (Tomy, Tokyo, Japan) at 780× *g* for 20 min at 22 °C, the interface between 25 and 40% gradients was saved to collect IECs. IECs were further enriched using EpCAM microbeads (Miltenyi Biotec, Gaithersburg, MD, USA) and magnetic-activated cell sorting (MACS) cell separation columns (Miltenyi Biotec). The enriched IECs were tested for EpCAM expression using a monoclonal antibody to EpCAM (G8.8) (eBioscience, San Diego, CA, USA), the rat IgG2a isotype control antibody (BioLegend, San Diego, CA, USA), and a BD Accuri C6 Flow Cytometer and BD Accuri C6 Software (BD Biosciences, San Jose, CA, USA).

### 2.4. IEC MicroRNA (miRNA) and Messenger RNA (mRNA) Analysis Using Deep Sequencing

RNA was extracted from IECs using a miRNeasy Mini Kit (Qiagen, Germantown, MD, USA). Library construction and sequencing of small RNAs (including miRNAs) were achieved by using an Ion Total RNA-Seq Kit v2 (Thermo Fisher Scientific, Waltham, MA, USA) and the Ion Personal Genome machine (PGM) system (Thermo Fisher Scientific) according to the manufacturer’s instructions at the Mie University Center for Molecular Biology and Genetics (Tsu, Japan) as previously described [30]. Data collection was performed with Torrent Suite v4.0.1 software. The assessment of miRNA profiling was conducted as previously described [30]. In brief, detectable miRNAs (>0, RPKM) across all samples were chosen for differential expression and downstream pathway analysis [32]. Individual fold changes (RPKM in CS-injected sepsis mouse/RPKM in sham mouse) were calculated by taking the ratio of the candidate miRNA expression values with one sham control. Those miRNAs and mRNAs with a fold change of 2 or greater (FC > 2) were classified as upregulated miRNAs and mRNAs in the sepsis group compared to sham, while those with (FC < −2) were classified as downregulated miRNAs and mRNAs in the sepsis group compared to sham.

### 2.5. Cell Culture and LPS Treatment

Human IEC lines (C2Bbe1, HUTU80, and H747) were obtained from ATCC (Manassas, VA, USA). The cells were cultured in RPMI 1640 supplemented with 10% fetal bovine serum (FBS) (Equitech-Bio, Kerrville, TX, USA) and penicillin (100 U/mL)/streptomycin (100 μg/mL) (Nacalai) in 5% CO_2_ at 37 °C. The cells of 70 to 80% confluency on a 6-well plate (Corning, Glendale, AZ, USA) were treated with lipopolysaccharide (LPS) (L3880, Sigma, St. Louis, MO, USA) at a concentration of 1 μg/mL and incubated for 24 h for further study, as described previously [33,34,35,36].

### 2.6. Real-Time Quantitative PCR (RT-qPCR)

RT-qPCR was performed as previously described [29,30,31] with slight modifications. Briefly, RNA was extracted from the cells and blood using a miRNeasy Kit (Qiagen, Hilden, Germany) and TRIzol reagent (Thermo Fisher Scientific) according to the manufacturers’ instructions. Approximately 1 μg RNA was subjected to a reaction of a reverse transcription using a Mir-X miRNA First-Strand Synthesis Kit (Takara Bio, Shiga, Japan) and a Prime Script RT Kit (Takara Bio), to detect the expressions of miRNAs and mRNAs, respectively. To examine relative gene expression, qPCR was conducted using a PowerUp SYBR Green Master Mix PCR kit (Applied Biosystems, Foster City, CA, USA) and the StepOne Real-Time PCR System (Applied Biosystems) according to manufacturer’s instructions. For endogenous controls, *U6* and *β-actin* were used to normalize expressions of miRNAs and mRNAs, respectively. For miRNAs, the universal primer (Thermo Fisher Scientific) was utilized as the reverse primer for miRNA validation runs. All the PCR-primer sequences for RT-qPCR used in this study are listed in Appendix A. Relative expression was calculated using the comparative threshold (CT) method (2^−dCT^) normalized to endogenous control genes and expressed between two cohorts.

### 2.7. MiRNA-Target Network and Pathway Analyses

The miRNAs of the IECs were applied to miRNet 2.0 [37], which incorporates miRBase [38], and miRTarBase v8.0 [39], to construct the networks. The Reactome, Gene Ontology, and KEGG analyses were performed in miRNet 2.0 [37].

### 2.8. Statistical Analysis

Data are presented as the mean ± standard error of the mean (SEM). Results were analyzed using two-tailed Student’s *t* test for comparison of two groups. *p*-values < 0.05 were considered significant. Statistical analysis was completed using Prism 8 software (GraphPad, San Diego, CA, USA).

## 3. Results

### 3.1. Sepsis Alters miRNAs and mRNAs in IECs in Mice

To examine the role of IEC-specific miRNAs and its downstream regulatory networks in sepsis, the CS injection model was used to induce sepsis in mice [27] and miRNA expression within IECs was investigated. The CS (25 mg of cecal contents resuspended in 10% glycerol in PBS) was injected intraperitoneally into each mouse, followed by antibiotic treatment (3 mg meropenem plus 3 mg cilastatin per dose; 7 doses). Sham mice received the same volume of 10% glycerol in PBS (0.25 mL per mouse) followed by treatment with antibiotics (Figure 1A). The survival rate was approximately 67% (12/18 mice) at day 17 after CS injection, while 100% survival (10/10 mice) was seen in shams.

At post-injection day 17, all mice were euthanized, the small intestines were removed, and IECs were isolated using Percoll density gradients [29,31] and further enriched using magnetic sorting with CD326 (epithelial cell adhesion molecule; EpCAM) microbeads (Figure 1B). After isolation of IECs, their exclusive expression of EpCAM was validated using flow cytometry (Appendix A). The whole transcriptome of small RNAs including miRNAs of the isolated IECs was sequenced using the high-throughput Ion Xpress™ RNA-Seq platform. Among 1076 miRNAs initially detected using the sequencing platform, 239 miRNAs that had any expression (i.e., threshold detection hit of >0) for all analyzed samples (1 sham and 3 septic mice) were analyzed and listed in Appendix A.

The average miRNA expression level in IECs after sepsis was compared with a sham counterpart with a threshold cutoff of fold change (FC) of 2 or greater. As shown in Figure 2A, the mean expression of 35 miRNAs was upregulated in IECs after sepsis compared with sham IECs, while 15 miRNAs were downregulated following sepsis. Figure 2A depicts miRNA candidates plotted across both mean fold change and mean expression (RPKM values). We further compared miRNA reads of each sample with the respective sham control for a more robust identification of miRNAs differentially expressed after sepsis. Following this filtering criteria, our data indicated that 14 miRNAs (miR-669o, miR-3096, miR-466q, miR-511, miR-495, miR-467e, miR-434, miR-154, miR-669a-4, miR-127, miR-328, miR-669a-5, miR-378c, and miR-149; ordered by FC) were upregulated in IECs after sepsis. Nine miRNAs (miR-6238, miR-1258, miR-124-2hg, miR-17hg, miR-5125, miR-6240, miR-351, miR-717, and miR-1983; ordered by FC) were downregulated in IECs after sepsis (Figure 2B).

To investigate any downstream regulating effects of the identified miRNAs, we analyzed the whole transcriptome and acquired comprehensive expression signatures of mRNAs using the same IEC samples of sham and sepsis mice as used in the miRNA analysis. We found that, among a total of 14,316 mRNAs detected, 2248 mRNAs were downregulated with sepsis, while 612 mRNAs in the IECs were upregulated (Figure 2C). Both up- and down-regulated mRNAs, shown as dots, were determined by the same criteria used in examining miRNAs for the mean RPKM values across the sepsis IECs. The number of downregulated mRNAs was approximately 3.6-times higher than that of the mRNAs upregulated by sepsis. Thus, this suggests that the miRNAs upregulated in IECs after sepsis contribute, at least partly, to the downregulation of the comprehensive mRNA profile.

We next sought to further identify the upstream regulating factors that affect the changes in gene expression of RNAs (including both mRNAs and miRNAs). DNA methylation is an epigenetic marker that effectively silences transcription [40] and requires enzymatic activity of DNA methyltransferases (e.g., DNMT1 and DNMT3A) [41,42]. We thus examined the expression levels of *DNMT1* and *DNMT3A* and found that both genes were significantly upregulated within the IECs of CS-injected sepsis mice compared to those of sham mice (Appendix A). Thus, these results suggest that sepsis induces an alteration of the overall transcriptome that may be related to epigenetic modifications. This can be achieved by the enzymatic activity of DNMT1 and DNMT3A in DNA methylations and/or by the posttranscriptional regulation of the miRNAs (such as miR-149-5p, miR-466q, miR-495, and miR-511-3p) upregulated in the sepsis IECs.

### 3.2. Sepsis-Upregulated miRNAs Provide a Highly Complex miRNA–mRNA Regulatory Network in IECs

To further elucidate the functional role of these IEC miRNAs in the pathophysiology of sepsis, we constructed miRNA–mRNA target interaction networks using an analytic platform, miRNet 2.0 [37]. As shown in Figure 3A, upregulated miRNAs in IECs after sepsis showed complex networks with putative targets. A large continent network encompassing multiple miRNAs, including miR-149-5p (423 targets), miR-495-3p (207 targets), miR-511-3p (130 targets), and miR-466q (105 targets), was identified as pivotal nodes. In a separate island network, miR-127-5p potentially targeted only one gene transcript. Taken together, these results indicated that IECs dynamically respond to sepsis by altering miRNA profiles and downstream sepsis-induced regulatory gene networks.

In contrast to the networks containing upregulated miRNAs, only two networks (one continent and one island) were identified which were regulated by the pool of downregulated miRNAs in IECs after sepsis (Figure 3B). The continent network incorporated 3 miRNAs including miR-5125 (194 targets), miR-717 (101 targets), and miR-1983 (20 targets). In contrast, miR-351-5p created 1 island network with 20 targets. However, 5 other miRNAs (i.e., miR-6238, miR-1258, miR-124-2hg, miR-17hg, and miR-6240) did not show any network interactions in the miRNet analysis platform.

### 3.3. In Silico Analysis Reveals Enriched miRNA-Regulated Pathways in IECs following Sepsis

To examine underlying pathways regulated by the candidate miRNAs in sepsis, we used Reactome [43], a high-performance bioinformatics tool, within miRNet 2.0. Following Reactome analysis, we identified a total of 100 pathways potentially regulated by the 14 upregulated miRNAs in IECs after sepsis, filtered by the significance level (adjusted *p* < 0.05) and then ranked by the number of hits (the number of gene targets involved in the given pathway; Appendix A). The top 20 pathways were listed by a 3-way bubble plot depicted using the number of hits, the significance level, and the gene ratio (Figure 4A). Interestingly, 6 out of the 20 pathways shown to be altered with sepsis were related to the fibroblast growth factor receptor (FGFR) signaling. Although not identified within the top 20 pathways, apoptosis and programmed cell death pathways were significantly enriched, indicating that sepsis may trigger epithelial cell death in the small intestine. In addition, gene ontology (GO) analysis revealed that a total of 99 pathways (adjusted *p* < 0.05) were identified (Appendix A) and the top 20 pathways (ranked by the number of hits) were shown in Figure 4B, depicted by the same criteria used in Reactome analysis. To further identify candidate pathways, we performed systemic enrichment analysis on the identified upregulated pool of IEC miRNAs with sepsis, using the Kyoto Encyclopedia of Genes and Genomes (KEGG) platform, a downstream pathway analysis tool within miRNet 2.0. The KEGG analysis showed a total of 57 pathways, filtered by the significance level (adjusted *p* < 0.05) and ranked by the number of hits (Appendix A). The top 20 pathways potentially regulated by miRNAs following sepsis were listed by a 3-way depiction, as described above (Figure 4C). Many of the notable pathways involved pathways in cancer. In addition, epithelial Wnt signaling in the small intestine is potentially altered by sepsis.

For the downregulated miRNAs seen in IECs after sepsis, we also performed Reactome, KEGG, and GO analyses. However, both the Reactome and GO analyses did not show any pathway. Only KEGG analysis revealed that Janus kinase-signal transducer and activator of transcription (JAK-STAT) signaling and ERBB signaling are putatively involved in the pathways regulated by the miRNAs downregulated in sepsis (e.g., miR-351-5p, miR-717, miR-1983, and miR-5125) (Appendix A). In conclusion, upregulated miRNAs may potentially be actively involved in cell-specific responses in the IECs after sepsis.

### 3.4. Validation of the miRNAs Altered in Sepsis IECs and Blood and miR-511-3p Emerged as a Diagnostic Marker

We next performed validation of the identified miRNA candidates. Most of the miRNAs upregulated in sepsis, including miR-149-5p, miR-154-3p, miR-328-3p, miR-378c, miR-434-5p, miR-466q, miR-467e-5p, miR-495-5p, miR-511-3p&-5p, and miR-699o-3p&-5p, were validated for their increased expression via quantitative PCR using RNA and complementary DNA of sepsis and control (sham) IECs (Figure 5 and Appendix A). These results suggest that sepsis produces a miRNA signature that is specific to the mucosal compartment and has the potential for the discovery of novel epithelium-specific biomarkers in sepsis.

To confirm that there were no off-target effects in sham mice that were injected with antibiotics (Figure 1), which were used as the control in this study, we included an additional cohort of mice which were not given antibiotics. To examine whether the levels of miRNAs increased in sepsis IECs are affected by the antibiotic injections alone, we assessed miRNA expression in IECs of the control sham mice in comparison with those of the mice with no injection (namely, normal mice). None of the 16 miRNAs that were upregulated in sepsis IECs showed any significant alterations in sham (control) mice injected with antibiotics when compared with those in uninjected normal mice (Appendix A). Consequently, these data indicate that injections with antibiotics did not affect the expression level of any of the sepsis-increased 16 miRNAs tested.

To examine if human cells exhibit a similar expression pattern of the miRNAs (including miR-149-5p, miR-466q, miR-495, and miR-511-3p) increased in the IECs of sepsis mice, we validated the levels of these miRNAs in human IEC lines, such as C2Bbe1, HUTU80, and H747. To provide an in vitro sepsis environment, we treated LPS to the cells, as described previously [33,34,35,36], and examined their miRNA expression levels. Cells treated with LPS did not show a significant change in the expression of the miRNAs upregulated in the IECs of sepsis mice (Appendix A). Thus, these results suggest that human IECs from cell lines may differ from sepsis-mouse IECs in miRNA expression in the currently used in vitro model of experimental sepsis. The profile of miRNAs in human IECs with sepsis requires further assessment using IECs isolated from the patients diagnosed with sepsis.

Next, we investigated the expression of the downregulated miRNAs (including miR-351-5p, miR-717, miR-1258-3p, miR-1983, and miR-5125) for validation and confirmed the reduction in miR-351-5p, miR-717, and miR-5125 expression levels in sepsis IECs by RT-qPCR. The expression of miR-1258-3p and miR-1983 did not show a significant downregulation in the IECs of the sepsis mice, compared to sham mice (Figure 6).

To ask if the expression patterns of up- and down-regulated miRNAs are restricted to sepsis IECs, we next investigated expression of the miRNAs in blood samples. The upregulated miRNAs selected for further study (miR-149-5p, miR-466q, miR-495-3p, miR-495-5p, and miR-511-3p) and the downregulated miRNAs (miR-351-5p, miR-717, miR-1258-3p, miR-1983, and miR-5125) were tested for their levels in blood of sham and sepsis mice. Among the 10 miRNAs (as shown in Figure 5 and Figure 6), only the miR-511-3p exhibited a significant augmentation, whereas the other miRNAs did not show significant changes (Figure 7). This suggests that this miRNA may be useful as a diagnostic marker in this sepsis model and should be examined in the blood of patients with sepsis in the future.

### 3.5. Reduced Expression of Putative Targets for the Sepsis-Increased miRNAs in IECs

We then explored gene expressions of specific mRNAs, such as *LOX*, *PTCH1*, *COL22A1*, *FOXO1*, and *HMGA2*, which were found to be bioinformatically targeted by one or more of the four identified miRNAs (miR-149-5p, miR-466q, miR-495, and miR-511-3p), as their gene products have been reported to associate with Wnt or inflammatory pathways. Specifically, the mRNA of the lysyl oxidase (LOX) is a putative target for miR-149-5p and miR-511-3p, and its downregulation activates the Wnt/β-catenin pathway [44,45]. The mRNA of the patched1 (*PTCH1*) is a putative target of miR-466q and miR-511-3p, and *PTCH1* targeting by miR-511-3p can activate the hedgehog pathway to trigger hepatic sinusoidal obstruction syndrome [46], which is promoted by inflammatory and fibrinolytic pathways. The mRNA of collagen type XXII α1 (*COL22A1*) is a putative target of miR-149-5p and miR-466q, and targeting of *COL22A1* by miR-149-5p regulates inflammation and fibrosis of cardiomyocytes [47]. The mRNA of Forkhead Box O1 (*FOXO1*) is a putative target for miR-466q, miR-495-5p, and miR-511-3p, and functional inhibition of *FOXO1* is associated with the Wnt/β-catenin pathway [48]. The mRNA of high mobility group A2 (*HMGA2*) was shown to be a target for miR-495 [49,50], and the *HMGA2* mediates the secretion of pro-inflammatory cytokines, while its downregulation induces hypermethylation [51]. Based on the literature and our current data that demonstrate a significant reduction in *LOX*, *PTCH1*, *COL22A1*, *FOXO1*, and *HMGA2* in the IECs of sepsis mice, compared to sham mice (Figure 8), the sepsis-augmented miRNAs in the IECs may have the potential for regulating both anti- and pro-inflammatory responses, possibly through posttranscriptional modification of their functional target genes. The proposed model illustrating miRNA-mediated target-gene expression regulations for anti- and pro-inflammatory responses in the IECs of sepsis mice is shown in Figure 9.

To analyze expression of the target genes such as *LOX*, *PTCH1*, *FOXO1*, and *HMGA2* in the human IEC lines subjected to LPS treatment, compared to control (no treatment), RT-qPCR was performed. Human cell lines (C2BBe1, HUTU80, and H747) treated with LPS showed no significant change in expression levels of the target genes (Appendix A) that were decreased in the IECs of the sepsis mice (Figure 8). Human IEC lines were expected to be largely different from those in mouse cells, with regards to expression of the miRNAs and their regulatory pathways, at least under the currently used in vitro model of experimental sepsis. Future studies are needed to evaluate miRNAs from human IECs in vivo and in the blood from septic patients.

## 4. Discussion

Here we have shown that sepsis induces distinctively expressed miRNA profiles in IECs in a mouse model of sepsis. The upregulated miRNAs exhibited more complex and broader effects on comprehensive gene regulations in silico related to both Wnt signaling and inflammatory pathways. These upregulated miRNAs in sepsis IECs may contribute to quantitative downregulation of overall mRNAs. Intriguingly, several distinct miRNAs (miR-149-5p, miR-466q, miR-495, and miR-511-3p) may suppress expressions of *LOX*, *PTCH1*, *COL22A1*, *FOXO1*, or *HMGA2*. Current findings could provide us insight into the miRNA–mRNA crosstalk in IECs that may contribute to pro- and anti-inflammatory responses in sepsis.

IECs are pivotal for the surveillance of intestinal environment to protect the host from both local and systemic challenges [52]. Disruption of the IEC barrier’s integrity caused by intestinal infection and inflammation has been shown to significantly shift the transcriptomic patterns within IECs [53]. Alterations of cell-specific epigenetic factors have received much attention as a potential contributor to the regulation of host mucosal immunosurveillance and IEC barrier function [54,55]. In this study, we aimed to uncover the epigenetic alterations to impact the disruption of the intestinal epithelium during sepsis by investigating the miRNA signature profiles.

miRNAs have been proposed as potential biomarkers for diagnosing sepsis [56]. Our data showed that sepsis upregulated the expression levels of 14 miRNAs in IECs, compared with sham counterparts. Bioinformatics analysis further revealed that miR-149-5p, miR-495, miR-511-3p, and miR-466q might be key epithelial miRNAs in the small intestines, following sepsis. Hūbner et al. found that miR-149-5p plays an important role in TLR-mediated inflammation of bronchial epithelial cells by directly regulating chitinase-3-like 1 (CHI3L1), which has been known to regulate the bacterial infection [57]. CHI3L1 is highly expressed in IECs and can contribute to bacterial adhesion and invasion in intestinal inflammation [58]. Further, Heinsbroek et al. demonstrated that miR-511-3p expressed by immune cells regulates microbiota-associated intestinal inflammation [59]. Thus, further investigation on the roles of miRNAs and biomarker discovery in the context of sepsis-induced IEC disruption is warranted.

In silico analysis further revealed that the upregulated miRNAs can regulate several pathways, including FGFR signaling. Al Alam et al. found that FGF and FGFR are expressed in both human and mouse small intestines [60]. In addition, FGF is significantly involved in cell differentiation of goblet cells and Paneth cells that are pivotal for epithelial protection in the intestines. Huang et al. also showed that inhibition of FGFR by a selective inhibitor, AZD4547, protected septic mice from pulmonary inflammation [61]. Song et al. suggested a protective role of FGF in a mouse model of intestinal inflammation [62]. More specifically, they found that FGF2 expressed by regulatory T cells cooperates with the cytokine IL-17 derived from Th17 cells to promote epithelial repair in a mouse model of intestinal inflammation [62]. Therefore, the interaction of FGFs and FGFRs in sepsis-induced intestinal epithelial inflammation and how miRNAs play a role as a mitigator of sepsis-induced inflammation are warranted for further investigation.

In the intestines, Wnt signaling is fundamental for epithelial homeostasis [63]. Wnt signaling regulates several cellular functions of IECs, such as intestinal stem cells, related to their capacities for self-renewal and differentiation [64]. Interestingly, our KEGG analysis suggests that epithelial Wnt signaling might be significantly regulated by miRNAs after sepsis. In addition, recent studies have demonstrated that Wnt signaling can be a therapeutic target for the regeneration of intestinal epithelium. Xie et al. showed that Wnt mimetics (molecules mimicking endogenous Wnt) can regenerate the damaged epithelial tissues and reduce inflammation in a mouse model of colitis [65]. Xu et al. also demonstrated that miRNAs are significantly involved in the activation of the Wnt pathway [66]. Indeed, a target prediction tool, TargetScanMouse 7.1, revealed that most of the upregulated miRNAs in IECs after sepsis can bind to the 3′-UTR region of the mRNAs of multiple Wnt genes. Thus, the role of intestinal epithelial miRNAs regulating Wnt signaling in tissue regeneration after sepsis is warranted.

Among downregulated miRNAs following sepsis, four miRNAs (i.e., miR-5125, miR-351-5p, miR-717, and miR-1983) displayed potential miRNA–mRNA networks in our analysis. There is little information on the pathways affected by the downregulated miRNAs seen in IECs after sepsis; one possibility is that those miRNAs are still comparatively novel and less documented in the literature. Although there has been a limited number of studies that demonstrate the role of those miRNAs, our KEGG analysis revealed that the JAK-STAT signaling pathway and ERBB signaling pathway might putatively be altered in SIECs after sepsis. Notably, our data showed that ERBB signaling is a common pathway which can be regulated by both upregulated and downregulated miRNAs. It has been shown that ErbB receptors and their ligands are crucial in epithelial cell recovery in mucosal tissues [67,68]. Therefore, further studies will investigate the involvement of ErbB signaling in the regulation of intestinal epithelial injury.

Dysregulation of IEC remodeling may lead to a sustained mucosal inflammatory response in sepsis. Impaired wound-healing and remodeling capacities in the IECs disrupt their barrier integrity and further lead to bacterial translocation and subsequent inflammation in the intestine and other systemic compartments. IECs dynamically respond to sepsis by altering their miRNA profiles to regulate both epithelial injury and regeneration.

Collectively, our data suggest that sepsis-induced inflammation is centralized toward persistent inflammation and immunosuppression. These findings are reminiscent of the spectrum of host responses typically seen in sepsis [69,70]. Further extensive understanding of sepsis-induced epigenetic alterations in different cell types, such as human IECs or leukocytes, would provide insight into the identification of therapeutic targets for sepsis.

## Figures and Tables

**Figure 1 cells-12-00726-f001:**
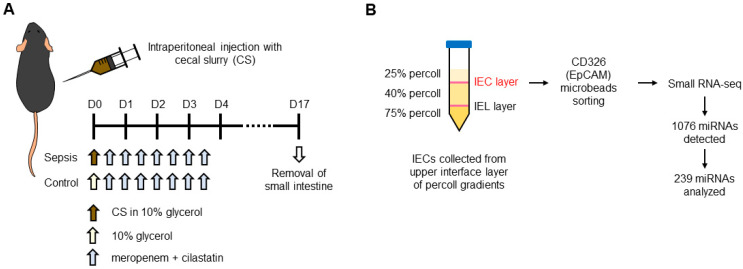
Experimental design of a mouse model of sepsis. (**A**) Mice were treated with cecal slurry (CS) by intraperitoneal injection to induce sepsis. CS was collected from naïve wild-type mice and dissolved in 10% glycerol in PBS. Sham mice were treated with 10% glycerol in PBS. After treatment, both sham and sepsis mice were treated with a cocktail of antibiotics (meropenem and cilastatin). At day 17 after treatment of CS, mice were euthanized, and small intestine tissues were collected for IEC isolation. (**B**) IECs were isolated from the small intestines by the methods of Percoll density gradients (25, 40 and 75%). Small intestinal tissues were separated from mice and Peyer’s patches, and fat tissues were then removed. The tissues were opened, washed with ice-cold PBS, and incubated with RPMI-1640 containing FCS (10%) and EDTA (2 mM). After filtering with 70-μm cell strainer, the filtered cell suspension was centrifuged, resuspended in 40% Percoll, and applied to Percoll density gradients. After centrifugation, IECs were collected from the interface between 25 and 40% of Percoll gradients. Isolated IECs were further enriched using magnetic sorting system with EpCAM or CD326 microbeads. MiRNAs of the isolated SIECs were sequenced using the high-throughput Ion Xpress™RNA-Seq platform. A total of 1076 miRNAs were detected, and 239 miRNAs that were detected across all samples (i.e., threshold detection hit of >0) were further analyzed for differential and downstream functional analysis.

**Figure 2 cells-12-00726-f002:**
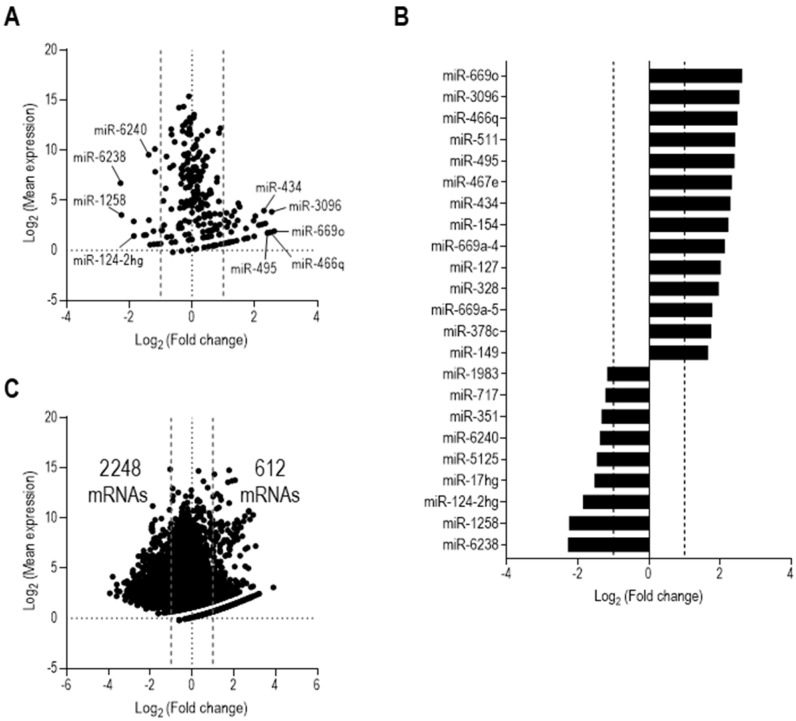
Sepsis significantly alters both miRNA and mRNA profiles in IECs. (**A**) Bioinformatic analysis of differentially expressed miRNAs in isolated small intestinal epithelial cells (IEC). A total of 239 miRNAs were identified following high-throughput miRNA-sequencing. 35 miRNAs candidates were identified with a positive fold change (FC of 2 or greater) on average, as identified by the data points in the right side of the plot denoting an upregulation with sepsis, while 15 miRNAs had a negative fold change (FC of −2 or lower) on average, as identified by the data points in the left side denoting a downregulation with sepsis. (**B**) Top differentially expressed miRNAs in IEC with sepsis. 14 differentially expressed miRNAs were upregulated with sepsis based on our selection criteria of fold change (FC of 2 or greater) for each sepsis sample compared to a sham control. Further, 9 differentially expressed miRNAs were seen to be downregulated with sepsis based on our selection criteria of fold change (FC of −2 or lower) for each sepsis sample compared to a sham control. (**C**) Whole transcriptome analysis of isolated IECs. A total of 14,316 mRNAs were identified following high-throughput mRNA-sequencing. 612 mRNA candidates were identified with a positive fold change (FC of 2 or greater) on average, as indicated by the data points in the right side of the plot indicating an upregulation with sepsis, while 2248 mRNAs had a negative fold change (FC of –2 or lower) on average, as denoted by the data points in the left side of the plot, denoting a downregulation with sepsis.

**Figure 3 cells-12-00726-f003:**
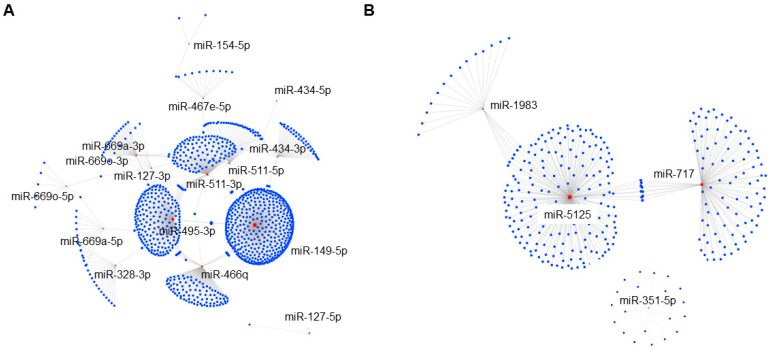
Sepsis-induced upregulated miRNAs exhibit more complex miRNA–mRNA regulatory networks in IECs. Putative miRNA–mRNA interaction networks of upregulated miRNAs (**A**) and downregulated miRNAs (**B**) in IECs with sepsis. Differentially expressed miRNAs and their predicted mRNA transcript targets are illustrated as networks across both upregulated and downregulated miRNAs with sepsis, in separate, as generated by an analytics platform, miRNet 2.0. Each individual node represents a miRNA (red) and an mRNA (blue), and grey lines signify a putative interaction. Relative sizes of miRNA and gene nodes depicted in the figure cannot be compared, as the interactive network analysis was run independently.

**Figure 4 cells-12-00726-f004:**
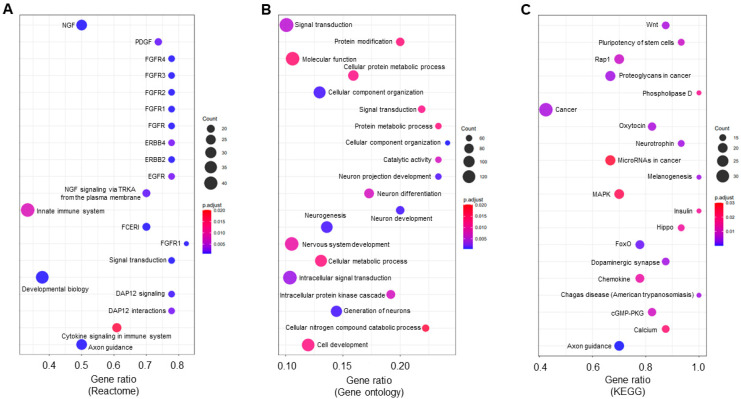
Bioinformatic analysis of downstream signaling and regulatory pathways by upregulated miRNAs with sepsis. Reactome (**A**), Gene Ontology (**B**), and KEGG (**C**) pathway analyses represented in three-way bubble plot analysis in the basis of gene ratio, gene count, and adjusted *p*-value. The top 20 pathways across the 3 downstream pathway analyses in IECs were regulated by 14 differentially expressed upregulated miRNAs with sepsis. The top 20 pathways across the 3 were selected by the rank of the number of hits. All pathways shown were filtered based on significant enrichment (adjusted *p*-value ≤ 0.05). Full names for pathways were provided in Appendix A.

**Figure 5 cells-12-00726-f005:**
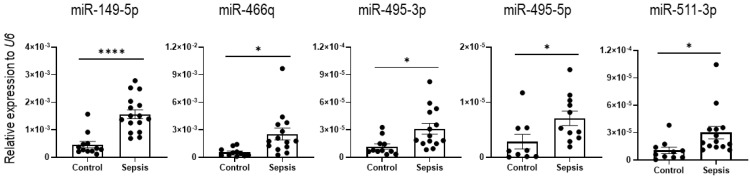
Validation of identified candidate miRNAs upregulated in sepsis IECs. The expression of upregulated miRNAs (miR-149-5p, miR-466q, miR-495-3p, miR-495-5p, and miR-511-3p) was tested using real-time quantitative PCR (RT-qPCR) analysis. *U6* was used as an endogenous control to normalize miRNA expression levels. Data are shown as scatter plots and bars overlaid with the mean ± standard error of the mean (SEM). Dot on the plot represents the value of each mouse. Control and sepsis indicate the IECs isolated from sham and sepsis cohorts, respectively. * *p* < 0.05 and **** *p* < 0.0001.

**Figure 6 cells-12-00726-f006:**
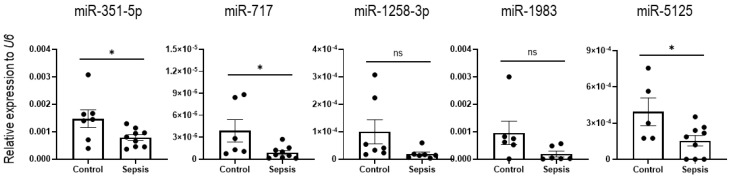
Validation of identified candidate miRNAs downregulated in sepsis IECs. The expression of downregulated miRNAs (miR-351-5p, miR-717, miR-1258-3p, miR-1983, and miR-5125) was tested by using RT-qPCR analysis. *U6* was used as an endogenous control to normalize miRNA expression levels. Data are shown as scatter plots and bars overlaid with the mean ± standard error of the mean (SEM). Dot on the plot represents the value of each mouse. Control and sepsis indicate the IECs isolated from sham and sepsis cohorts, respectively. * *p* < 0.05 and ns, not significant.

**Figure 7 cells-12-00726-f007:**
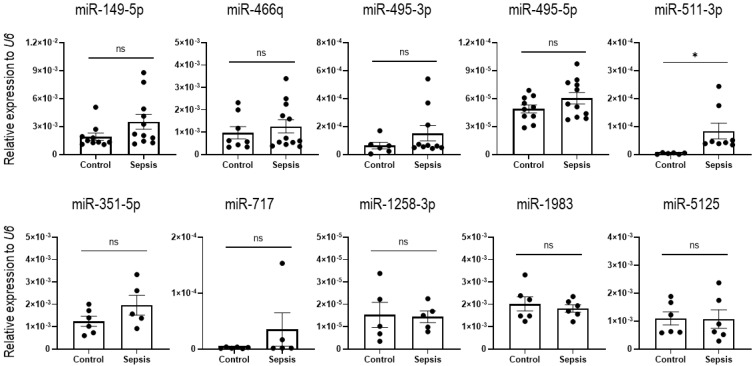
Validation of identified candidate miRNAs in blood. The expression of the miRNAs upregulated (upper) and downregulated (lower graphs) was analyzed with RNA extracted from the blood samples of sham and sepsis mice by using RT-qPCR analysis. *U6* was used as an endogenous control to normalize miRNA expression levels. Data are shown as scatter plots and bars overlaid with the mean ± standard error of the mean (SEM). Dot on the plot represents the value of each mouse. Control and sepsis indicate the blood samples drawn from sham and sepsis cohorts, respectively. * *p* < 0.05 and ns, not significant.

**Figure 8 cells-12-00726-f008:**
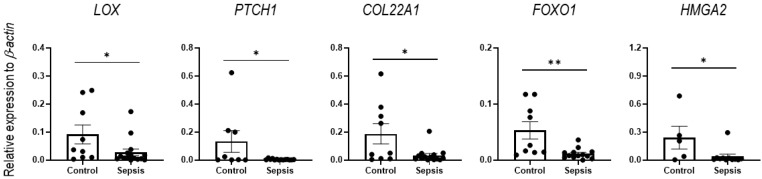
Reduced expression of target mRNAs in sepsis IECs. The expression of the mRNAs of putative targets (*LOX*, *PTCH1*, *COL22A1*, *FOXO1*, and *HMGA2*) was analyzed by using RT-qPCR analysis with RNA extracted from the IECs of sham and sepsis mice. *β-actin* was used as an endogenous control to normalize mRNA expression levels. Data are shown as scatter plots and bars overlaid with the mean ± standard error of the mean (SEM). Dot on the plot represents the value of each mouse. Control and sepsis indicate the IECs isolated from sham and sepsis cohorts, respectively. * *p* < 0.05 and ** *p* < 0.01.

**Figure 9 cells-12-00726-f009:**
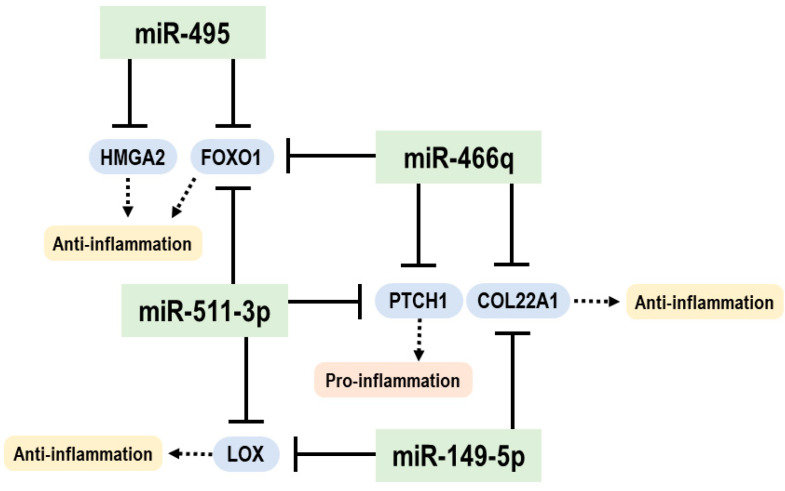
Graphical summary for a hypothetical model proposed in this study. A pool of distinct miRNAs such as miR-149-5p, miR-466q, miR-495, and miR-511-3p upregulated in the IEC of the sepsis mice may play roles in mediating anti- and pro-inflammatory responses by suppressing expression of their putative targets (e.g., *LOX*, *COL22A1*, *PTCH1*, *FOXO1*, and *HMGA2*).

## Data Availability

Data are contained within in the article or Appendix A and may be accessible upon reasonable request to E.J.P.

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
