# Peer review of "MicroRNA Profiles in Intestinal Epithelial Cells in a Mouse Model of Sepsis"

_cells, 2023, doi:10.3390/cells12050726_

Round 1

Reviewer 1 Report (Previous Reviewer 1)

The authors have provided supplementary experiments and a detailed explanation of the relevant issues, and I have no other comments.

Author Response

Thank you.

Reviewer 2 Report (New Reviewer)

very interesting manuscript. The authors should explain why they used that type of miRNA. it is recommended to define the role of miRNAs in the diagnosis of sepsis. It is requested to add this work to the bibliography: DOI: 10.3390/diagnostics11010032. We need to revise the English because there are some grammatical errors.

Author Response

In accordance with the suggestion of the reviewer, we have included the citation (ref. 56) with the description (line 533) in the revised manuscript. In addition, we have extensively revised the English with help of native experts. The detailed revisions are shown as track changes in red in the revised manuscript. Thank you.

This manuscript is a resubmission of an earlier submission. The following is a list of the peer review reports and author responses from that submission.

Round 1

Reviewer 1 Report

The study analyzed the expression profile of microRNAs (miRNAs) in IECs isolated from a mouse model of sepsis generated via cecal slurry injection. And then the authors did bioinformatic analysis with the miRNA data. In general, it is a very preliminary study. Study on miRNA profiles in IECs of human with sepsis is much more important than mouse model. To make the study acceptable for the journal, at least the following experiments should be done.

1.Analyze the upstream regulating factors of miRNAs such as methylation of certain DNA region by genome-wide analysis of methylation of the IECs and the downstream regulating effects of miRNAs such as downregulating mRNA level by RNA-seq analysis of mRNAs.

2.Validate the identified miRNAs in IECs of patients with sepsis or human cell lines with mimic sepsis environment.

3.Validate the identified pathways in IECs of patients with sepsis or human cell lines with mimic sepsis environment. 

4.Whether short-term continuous 6-day injection of antibiotics may affect the level expression of miRNAs in IECs in the sham mice. This may require more experimental results to prove.

5.The author used the cecal slurry by intraperitoneal injection to induce chronic-sepsis mice model, however, the modeling time was as long as 17 days and only 3 experimental mice were available, by what observation did the modeling prove successful? 

6.Please provide more information about the PCR primer sequences in this study in order to be able to repeat the experiments.

7.A total of 3 septic mice that were injected intraperitoneally with CS were included in the study, but only 2 mice remained after 17 days of injection. The number of samples in this study is too small, and a large sample study should be added for verification.

8.The author found the top 20 pathways across the 3 downstream pathway analyses in IECs were regulated by 14 differentially expressed upregulated miRNAs with sepsis but bioinformatic analysis analyses did not show any pathway for the downregulated miRNAs seen in IECs after sepsis. Why did these results occur? It is suggested that the author increase the possible mechanism analysis. This is more conducive to support the conclusions of this study.

Author Response

We would like to thank the reviewer for the valuable comments and suggestions on our manuscript, which have helped us improve it. We include a point-by-point response to the comments below. We have provided detailed revisions to address the issues raised in the initial review. These are shown highlighted in yellow in the revised manuscript.

Reviewer 2 Report

The manuscript describes the identification of dysregulated miRNAs in mouse model of sepsis. The disturbance of posttranscriptional regulation mechanisms in IECs is highly important in sepsis. The authors identify several miRNAs and propose models of dysregulated miRNA -mRNA networks leading to pro- and anti-inflammatory effects in sepsis.

Major concerns

Line 182 : “(one sham and three septic mice)”

To perform studies upon posttranscriptional regulation networks in mice, especially when networks regulating pro- and anti-inflammatory effects in sepsis are studied, at least 5 mice for each group should be used

The authors do not show a volcano plot of differential expression of the miRNA-seq data. Figure 2A shows is Log2(mean expression) vs log2fold change. It is not clear why they choose this visualization method since they already describe cutoffs in the manuscript and have not shown any statistical significance of the miRNA-seq data.

The models proposed within the manuscript depend on Putative miRNA-mRNA interaction networks and no effort was made to validate any of the pathways proposed. The author could have quantified several targets or perform RNA-Seq (since they have the material) and integrate miRNA dysregulation with mRNA levels genome wide. The authors also discuss the potential importance miR-149-5p on CHI3L1 and could use it as a model but have not performed experiments.

Minor points

Lines 198-199:  “The downregulated miRNAs were not further examined due to their shortage of information registered  in the analytic platform used (miRNet 2.0).”

The validation of these downregulated miRNAs should be performed for Data reliability purposes

The bar graphs should show individual points for each mouse (each point should be the mean of the technical replicate the number of which should be noted in the text or figure legend)

Author Response

(The authors gave the same response as above.)
